# Optimizing Betalactam Clinical Response by Using a Continuous Infusion: A Comprehensive Review

**DOI:** 10.3390/antibiotics12061052

**Published:** 2023-06-15

**Authors:** Sylvain Diamantis, Catherine Chakvetadze, Astrid de Pontfarcy, Matta Matta

**Affiliations:** 1Infectious Diseases Unit, Groupe Hospitalier Sud Ile de France, 77000 Melun, France; 2DYNAMIC Research Unit, Université Paris-Est-Creteil, 94320 Thiais, France

**Keywords:** betalactam, continuous infusion, intermittent infusion, pharmacoeconomics, clinical cure

## Abstract

Introduction: Antimicrobial resistance is a major healthcare issue responsible for a large number of deaths. Many reviews identified that PKPD data are in favor of the use of continuous infusion, and we wanted to review clinical data results in order to optimize our clinical practice. Methodology: We reviewed Medline for existing literature comparing continuous or extended infusion to intermittent infusion of betalactams. Results: In clinical studies, continuous infusion is as good as intermittent infusion. In the subset group of critically ill patients or those with an infection due to an organism with high MIC, a continuous infusion was associated with better clinical response. Conclusions: Clinical data appear to confirm those of PK/PD to use a continuous infusion in severely ill patients or those infected by an organism with an elevated MIC, as it is associated with higher survival rates. In other cases, it may allow for a decrease in antibiotic daily dosage, thereby contributing to a decrease in overall costs.

## 1. Introduction

Antimicrobial resistance is a growing concern around the world, and by 2050 may be responsible for up to 10 million deaths per year [1]. Along with antimicrobial resistance, inappropriate therapy is a leading cause of increased treatment failure, mainly in severe sepsis and septic shock [2]. Resistant organisms are, thus, a significant challenge to treat. Using broad-spectrum antibiotics could be one solution; however, it may lead to an increase in antimicrobial resistance [3]. A creative strategy would be optimizing the pharmacokinetic/pharmacodynamics (PK/PD) properties of the antimicrobials by using continuous perfusion of time-dependent antibiotics with low ecological impact in order to balance the continuous rise of bacterial MICs towards old molecules [4,5]. 

Betalactams are the most frequently prescribed antibiotics. They represent more than 65% of all antibiotics prescribed in the USA [6]. From a biochemical point of view, all betalactams have a common 3-carbons and 1-nitrogen ring (betalactam ring) that is highly reactive [7]. With the exception of penicillin V and G, most of them are hydrophilic [8]. Betalactams are comprised of four different major classes that are used in clinical practice. The first class is penicillin, discovered in 1929 [9], followed by cephalosporin, whose structure was first identified in 1961 [10], as well as carbapenems and monobactams. The latter two classes have been discovered more recently [11,12].

The parameter associated most frequently with clinical success when considering a betalactam-based therapy is the percentage of time with a free plasma concentration of the antimicrobial above the MIC (fT > MIC) [13]. Therapeutical drug monitoring is one way to ensure adequate concentrations; however, it is not readily available in all hospitals [14]. Another way is to adopt a prescription protocol that takes into account the PK/PD by optimizing the fT > MIC ratio, such as using a continuous infusion (CI) rather than an intermittent infusion (II) of betalactams, as CI is associated with a higher fT > MIC ratio [15], although it does not always translate into a better clinical outcome. Another point to take into consideration is the emergence of resistance, and regimens must provide drug concentrations that minimize the development of resistant mutants [16]. It has already been established that a CI of betalactam achieved better target attainment than an II [17].

Our primary objective is to review the evidence behind the use of CI of betalactams in clinical settings in terms of better clinical outcomes. Secondary objectives include the pharmaco-economic impact of a CI as well as its impact on the selection of resistance.

## 2. Methodology 

We reviewed published papers focusing on continuous infusion of betalactams in a clinical setting compared to an intermittent infusion. PubMed and Cochrane databases were searched with February 2023 as the publication date limit using MESH terms. The MESH terms included ((betalactam) OR (cephalosporin) OR (carbapenem) OR (penicillin)) AND ((continuous infusion) OR (intermittent infusion)). All clinical articles found by the automatic search and written after 2000 had their abstract reviewed by two authors (M.M. and S.D.). In case of disagreement, a third author (ADP) decided if the article should be included in the review. Review articles found in the search, as well as the references of the selected articles, were also reviewed for additional articles. The type of studies included were cohort studies, prospective open-label studies, and randomized controlled trials (RCT). In total, 1276 articles were found on PubMed and a further 75 on the Cochrane database. Further studies were added by review of the references. We also added separately the terms resistance and pharmacoeconomics (298 and 18 articles, respectively). These were reviewed using the same methods by the same authors. A total of 33 studies were included in the review of the main objective [18,19,20,21,22,23,24,25,26,27,28,29,30,31,32,33,34,35,36,37,38,39,40,41,42,43,44,45,46,47,48,49,50] (Table 1). Further 10 were included for the review of the pharmaco-economic impact [22,23,26,27,39,40,43,51,52,53] and 6 for the review of the impact on the resistance [29,38,54,55,56,57] (Figure 1).

## 3. Penicillins

Besides piperacillin/tazobactam, there are few studies that evaluated other antibiotics, such as cloxacillin [19], piperacillin [20], and temocillin [21].

### 3.1. Oxacillin

In a retrospective study comparing CI to II of oxacillin in patients with methicillin-susceptible Staphylococcus aureus (MSSA) endocarditis, Hughes et al. found that there was no difference in 30-day mortality (8% vs. 10%, *p* = 0.7) or length of stay (LOS) (20 versus 25 days, *p* = 0.4). The CI group had a higher microbiological cure at day 30 (94% versus 79%, *p* = 0.03). In multivariate analysis, the antibiotic regimen (CI vs. II) was the only significant variable associated with a better bacteriological cure (*p* = 0.04) [19].

### 3.2. Piperacillin

Rafati et al. compared the efficacy of a continuous infusion of piperacillin in relation to the more traditional intermittent one. They found that in critically ill patients, changes in APACHE II scores from baselines were significantly better on days 2, 3, and 4 in patients who received the CI when compared to those who received an II (*p* < or = 0.04). They also found that for organisms with a MIC of 16 or 32 µg/mL, the fT > MIC of piperacillin was significantly higher in the CI group. They concluded that the improved pharmacodynamics of the CI was likely the cause of the improved clinical result [20]. 

### 3.3. Temocillin

Laterre compared a CI of 6g temocillin over 24 h vs. an II of 2 g q8 h. The overall clinical cure rate was 84% (27/32); in the II and CI groups, it was 79% and 93%, respectively, favoring the CI. PK/PD analysis was in favor of the CI, and although the difference in clinical response showed a not statistically significant trend in favor of a CI despite a limited number of patients, it would be interesting to have a larger study powered to detect such difference [20]. 

### 3.4. Piperacillin/Tazobactam

Piperacillin/tazobactam was the most evaluated penicillin, with more than 10 studies [21,22,23,24,25,26,27,28,29,30,31,32,33]. These studies included different dosages and different populations. Primary and secondary endpoints were also different; therefore, comparing all these studies on a head-to-head basis is complicated. In 176 episodes of febrile neutropenia in hemato-oncological pediatric patients, Solórzano-Santos et al. did not find any difference in clinical improvement or treatment failure (21% in the CI group versus 13% in the II group (*p* = 0.15)) [21]. Likewise, Cotrina-Luque et al. compared a CI of piperacillin/tazobactam given as an II in hospitalized patients with suspected or confirmed pseudomonas. They found that the cure rate was comparable between the two arms of treatment. There were no adverse events in both arms [22]. Grant et al. compared various dosages of piperacillin/tazobactam in hospitalized patients. Both clinical and microbiological success rates were non significantly higher in the CI group compared to the II group: 94% and 89% vs. 82% and 73%, respectively. However, patients in the CI group had significantly fewer days with fever when compared to the II group (1.2 ± 0.8 days vs. 2.4 ± 1.5 days) (*p* = 0.012) [23]. In immunocompetent patients with complicated intra-abdominal infections and a median APACHE II score of 7, Lau et al. found that the continuous infusion was not associated with inferior clinical or bacteriological success [24]. In an RCT of critically ill patients in 25 intensive care units (ICUs) with a median APACHE II score of 20, Dulhunty et al. found that with a continuous infusion of broad-spectrum antibiotics, there were no differences in the clinical or microbiological outcomes. In this study, 26% of the patients were on renal replacement therapy, and patients received a CI treatment for a short duration frequently in combination therapy [25]. Finally, a small study by Buck et al. on 20 patients using 30% less in continuous infusion and another study by DeRycke also did not find any difference; however, the latter seems to include the same patients as Lau [26,27]. 

On the other hand, several other studies concluded a superiority of the CI, mostly in severe cases or when the infection is caused by a less susceptible organism. In a historical cohort study, Lorente et al. showed that in patients with ventilator-associated pneumonia, the CI of piperacillin/tazobactam was significantly better than an II regarding clinical cure. Cure rates were 88.9% when a CI was used vs. 40.0% when piperacillin/tazobactam was administered as an II (odds ratio (OR) = 10.79, 95% confidence interval (CI), 1.01–588.24; *p* = 0.049) when the causative organism had a MIC of at least 8 mg/L. For patients infected with microorganisms having a MIC of 16 mg/L, cure rates were 87.5% as CI vs. 16.7% as II (OR = 22.89, 95% CI, 1.19–1880.78; *p* = 0.03) [28]. In another recent retrospective study of critically ill patients, Hyun found that the 28-day mortality rates were significantly lower in the CI group even after adjusting for covariables (12.8% vs. 27.3%; adjusted hazard ratio (HR), 0.31; 95% confidence interval (CI), 0.13–0.79; *p* = 0.013). The CI group also had a higher probability of being ventilator-free at 14 days than the II group (HR, 1.77; 95% CI, 1.10–2.84; *p* = 0.018) and a higher rate of being discharged alive from the ICU at 14 days (HR, 1.95; 95% CI, 1.27–3.01; *p* = 0.002). These two parameters were similar between the two groups on day 28 [29]. Two prospective studies studied the impact of a CI in patients with pneumonia. In an open-label randomized study, Li et al. found that clinical was significantly better when a continuous infusion was used (75.0% vs. 50.0%; OR, 3; 95% CI, 1.05–8.53; *p* = 0.04); there was no statistically significant difference in the microbiological cure rate [30]. The same result was achieved by Abdul-Aziz [a] in a post hoc analysis of a prospective pharmacokinetic post-prevalence study, wherein with the subgroup of patients with respiratory infections, a continuous infusion was associated with a significantly better 30 days [86.2% (25/29) versus 56.7% (17/30); *p* = 0.012] [31]. In the same study, patients with a SOFA score >9 also had a better outcome when continuous infusion was used. In two other RCTs in ICU, with 60 and 140 critically ill patients, respectively, both studies had a higher clinical cure rate (70% vs. 43%, *p* = 0.037 and 56% versus 34 %, *p* = 0.01). There were no significant differences in survival rates [32,33].

## 4. Cephalosporins

Besides one study concerning ceftriaxone, all other studies evaluated ceftazidime or cefepime. Most studies compared an extended infusion to an intermittent one; however, there are few that evaluated a CI regimen.

### 4.1. Ceftriaxone

Roberts et al. evaluated a CI of 2 g of ceftriaxone to a once-daily infusion of the same dosage in ICU patients admitted with sepsis. Although the intention to treat analysis (57 patients) did not find a clinically significant difference in clinical cure [CI *n* = 13/29 versus II *n* = 5/28; adjusted odds ratio (AOR), 3.74; 95% CI, 1.11–12.57; *p* = 0.06], in the analysis of the population (50 patients) who completed at least 4 days of treatment (the a priori definition), CI was associated with a better outcome after adjusting for the age and the SOFA score when compared (AOR = 22.8; 95% CI = 2.24–232.3; *p* = 0.008) [34].

### 4.2. Ceftazidime

The clinical efficacy of ceftazidime is less established. Several studies compared the impact of a CI of ceftazidime in nosocomial pneumonia. There was no difference between the CI and II in terms of efficacy and adverse events in a study comparing 3 g daily as CI to 2 g q8h II for nosocomial pneumonia, despite the use of 50% lower doses in the CI [35]. The prospective study by Nicolau et al., also on patients with nosocomial pneumonia, did not find any difference, having also used the same discrepancy in dosage between the two infusion methods [36]. Both studies included the same set of patients. A third study by Hanes et al. also did not find any differences in terms of clinical response between the two methods of infusion in nosocomial pneumonia in trauma patients. *Haemophilus influenza* was the most frequently identified microorganism, with a mean MIC of 0.55 mg/L [37]. A retrospective study of 121 patients with ventilator-associated pneumonia, with a mean APACHE ll score of 16.08, showed that upon logistic regression analysis, CI was associated with a greater clinical cure rate than intermittent infusion (89.3% vs. 52.3%; OR, 12.2; 95% CI, 3.47–43.21; *p* < 0.001) [38].

In patients with cystic fibrosis, three studies compared a CI of ceftazidime to an II and did not find any advantage for a CI regimen in terms of clinical cure frequency. In a prospective study, Rappaz et al. showed that 100 mg/kg of ceftazidime as a CI was as effective as a dosage of 200 mg/kg given as an II. However, PK/PD favored the CI, as 32% of blood samples in the II group had a through concentration of ceftazidime below the MIC of the *Pseudomonas aeruginosa* isolated in sputum, while in the CI group, there was no ceftazidime concentration in plasma below the MIC [39]. Hubert et al. conducted a crossover study among children affected by cystic fibrosis. He found that a CI of ceftazidime was as efficient as an intermittent infusion regimen. The time between relapses was longer in patients receiving a CI (3.2 months) than in those receiving II (2.8 months; *p* = 0.04). There was no difference in quality-of-life scores or reported side effects [40]. Finally, Riethmueller found that in a randomized crossover study comparing a 50% less CI vs. an II (100 mg/kg in CI compared to a 200 mg/kg in II), both infusion methods were equally effective in improving lung function parameters and lowering the inflammatory markers such as leucocytes and C reactive protein value as well as the bacterial load in the airways [41].

### 4.3. Cefepime

Few studies evaluated a CI of cefepime compared to an II. In neurosurgical patients with postoperative intracranial infection, although CI controlled intracranial infection more rapidly and effectively than II, there was no difference in the rate of the clinical cure. PK/PD parameters were in favor of the CI as steady state concentration (Css) was greater than 4 times the MIC of 8 mg/L during a CI, while for the II regimen, mean serum cefepime concentration was above 8 mg/L for 81.66% of the dosing interval. They concluded that both methods can be used as an effective mode [42]. Another prospective RCT including critically ill patients, the BLISS study, found that the CI of betalactam, including cefepime, was associated with a better clinical cure and more mechanical ventilation-free days at day 28 (56% versus 34 %, *p* = 0.011 and 22 versus 14 days, *p* = 0.043) [32]. In another prospective randomized parallel study comparing a continuous infusion of cefepime vs. an intermittent regimen in 50 critically ill adult patients with Gram-negative bacilli pneumonia or bacteremia, it was found that the continuous regimen had a better PK/PD profile allowing for a greater bactericidal activity even in the absence of a difference in the clinical outcome [43]. Finally, in cystic fibrosis patients, a continuous infusion of cefepime of 100 mg/kg/day compared to an intermittent infusion of 50 mg/kg q8 h was associated with a greater although non-significant improvement in the pulmonary function tests despite the use of a lower overall dose. They also demonstrated that the CI allowed them to attain better pharmacodynamic targets while using a lower dose [44]. 

## 5. Carbapenems

Most studies comparing a continuous infusion of a carbapenem to an intermittent one involved meropenem. 

### 5.1. Imipenem

An exception to this was the study by Sakka et al., who published a small RCT comparing a continuous vs. a short-term infusion of imipenem in 20 patients. Two patients died in the intermittent group versus one patient in the continuous group, but the infusion method was not associated with mortality in the covariate analysis. This study was not powered for detecting a difference in mortality, and CI was prescribed for only 72 h with a lower overall dose compared to an II. The MIC of the responsible organisms were low, with a maximum of 0.5 mg/L in the II vs. 2 mg/L in the CI. However, Monte Carlo simulations showed that a CI was associated with a better PTA, notably for organism with a MIC > 2 mg/L [45].

### 5.2. Meropenem

For meropenem, different studies showed contrasting results. In a study including patients with moderate community-acquired pneumonia in the elderly, Okimoto compared a 1g of meropenem in a CI to a 0.5 g q12 h as an II. He did not find any difference in clinical response or duration of treatment; however, it is likely that because of the study design, there were no patients sick enough nor ones infected with resistant microorganisms in order to detect any difference [46]. The same conclusion was reached by Dulhunty et al. In a multicentric study including 432 patients with severe sepsis in 25 ICUs, they did not find any difference in ICU-free days, clinical cure, or 90-day mortality rate [25]. On the other hand, a study by Lorente et al. including critically ill patients with ventilator-associated pneumonia, found that a continuous infusion of 4 g of meropenem was significantly better than an II of 1 g q6 h in terms of clinical cure rate with a rate of 90.47% for the CI group vs. 59.57% for the II group (OR 6.44 [95% Cl 1.97 to 21.05; *p* < 0.001]) [47]. Zhao et al. also compared a CI of meropenem to an II in patients admitted to the ICU with severe sepsis or septic shock. They found that the CI was associated with a shorter duration of treatment (7.6 vs. 9.4 days; *p* = 0.035), with a better, although non-significant, microbiological cure rate (81.8% vs. 66.7%) [48]. Chytra et al. found that although the clinical cure rate was comparable between the CI and II groups (83.0% vs. 75.0%; *p* = 0.180), the microbiological success rate was higher in the CI group (90.6% vs. 78.4%; *p* = 0.020). Multivariate logistic regression identified the CI of meropenem as an independent predictor of microbiological success (OR = 2.977; 95% CI = 1.050 to 8.443; *p* = 0.040). Furthermore, meropenem-related ICU stay was shorter in the CI group (10 days vs. 12 days; *p* = 0.044). Meropenem therapy duration was also shorter in the CI group (7 days vs. 8 days; *p* = 0.035) [49]. Three other RCTs set in ICU with critically ill patients, including 211, 140, and 90 patients, have also compared the two infusion methods when prescribing a betalactam (meropenem or piperacillin/tazobactam or ticarcillin/clavulanic acid) [31,32,33]. The first by Abdul-Aziz et al. found that the continuous infusion was associated with a higher clinical cure rate (73.3% (11/15) versus 35.0% (7/20); *p* = 0.035) and better survival rates (73.3% (11/15) versus 25.0% (5/20); *p* = 0.025) in patients with a SOFA score of ≥9 and in patients with pneumonia (86.2% (25/29) versus 56.7% (17/30); *p* = 0.012) [31]. The second study, also by Abdul-Aziz, found that in the CI group, patients also had a higher clinical cure rate (56 versus 34 %, *p* = 0.011) and more median ventilator-free days (22 versus 14 days, *p* = 0.043) than patients in the II group [32]. Dulhunty et al. found that a CI was associated with a significantly higher clinical cure rate when compared with an II (70% vs. 43%; *p* = 0.037), while ICU-free days (19.5 vs. 17 days; *p* = 0.14) and survival rate (90% versus 80%, *p* = 0.47) were not significantly different [33]. Finally, in a 2015 study of patients admitted to the ICU with severe sepsis, Helmy et al. found that a continuous infusion was associated with a significantly shorter ICU stay, a lower SOFA score, and WBC at day 5. Mortality was not significantly different in the study (26% in the CI vs. 38% in the II *p* = 0.198) [50].

## 6. Pharmaco-Economic Impact

Several studies evaluated the pharmaco-economic impact of a continuous infusion. 

De Ryke evaluated the pharmaco-economic impact of a CI of piperacillin/tazobactam. He did not find any difference in level 1, 2, or 3 costs between the two regimens. In the CI group, there were lower labor and supply costs (*p* < 0.001), but overall costs were the same [27]. Other studies evaluated a lower dose of CI compared to a higher dose of an II and found significant savings. Grant et al. found that a lower daily piperacillin/tazobactam dose of 9 g administered via CI had lower level 2 costs when compared with an II of 13.5 g/day (399.39 ± 407.22 vs. 523.49 ± 526.85 *p* =0.028) [23]. Likewise, Buck et al. demonstrated that a 9 g CI piperacillin/tazobactam provided antibacterial activity similar to that of an II regimen, with a 15% reduction in the mean daily dose [26]. Florea et al. also found that compared to an II, a CI of piperacillin/tazobactam was more cost-efficient on a daily level as well as either as a 5- or 10-day regimen [51]. Finally, Kotopati and colleagues found that a CI of piperacillin/tazobactam had numerous advantages, including nursing labor cost [52], and Cotrina-Luque found that a CI could lead to an administration of a 30% less dose of piperacillin/tazobactam [22]. 

In patients with cystic fibrosis, several studies indirectly showed a better pharmaco-economic profile for a CI. Several studies showed that a CI achieves the targeted serum concentration while using a lower dose of antibiotics (up to 50% less for ceftazidime and about 17% less in the case of cefepime) therefore leading to lower costs [39,44,53]. Furthermore, as stated before, a CI was associated with a lower rate of exacerbations which translated into fewer hospitalizations and, therefore, lower overall costs [40].

## 7. Impact on Resistance

Few studies evaluated the effect of using a CI of betalactam instead of an II on the selection of resistant microorganisms. Most did not find any resistant organism breakthrough in both arms [30,39]. Felton et al. also compared a bolus infusion over 30 min to an extended infusion over 4 hours of piperacillin–tazobactam using a hollow fiber infection model. They did not find any difference in terms of the emergence of bacterial resistance [54]. Similar results were published by Dhaese et al., who found that for patients treated with piperacillin/tazobactam or meropenem, the mode of infusion was not associated with the emergence of resistance [55]. However, a study by Tamma et al. found that the use of an extended infusion of ceftolozane–tazobactam was protective against the emergence of resistance when compared to an II (0% vs. 29% *p* = 0.04) [56]. Finally, in a non-comparative study, Gatti et al. showed that when a critically ill patient had the Css of the antibiotic infused less than five times the MIC of the treated bacteria or an infection due to a Pseudomonas aeruginosa, he had a higher chance to select a resistant organism [57].

## 8. Discussion

The studies comparing a CI of betalactam to an II are very heterogeneous. They did not use the same dosages nor did they include the same population. Some studies included only documented infections [42], while in others, the proportion of microbiological identification was low [21,22,25,45]. When identified, the median MIC was different between the multiple studies. In some, MIC was always below 0.5 mg/L, while in others, it was as high as 16 mg/L or even 32 mg/L [20,28,45]; this difference in MIC could be responsible for the different results between the multiple studies. Another confounding factor is that some studies used an association of multiple antibiotics, adding a source of bias to the results [26]. The population included in the studies was also small in many studies, thereby limiting the value of the results as the studies were not powerful enough to detect a change in mortality [26,27]. Some studies with a low population detected a difference in mortality; however, it was not at a significant level, and it makes us wonder if different results would have been found if more patients had been included [24,50]. Some studies also had major limitations. In one of the studies, 81% did not have any clinical or microbiological diagnosis [21,25]. Fever also resolved in less than 48 h in almost half the febrile neutropenia cases hinting at a non-infectious etiology [21]. In several others [19,26,35,36,39,41,49], a lesser dose of a CI was compared to a higher one. This difference in dosing is a major bias that could have influenced the clinical results. Another limitation is that very few studies have included patients with infections at a site where antibiotic penetration is difficult therefore requiring a high-dose antibiotic regimen such as osteomyelitis, endocarditis prosthetic joints, or vascular graft infections. A final limitation is that some studies could have been missed because of the language of the study. We also did not compare a CI to an extended infusion.

Despite these limitations, some trends could be deduced. When the population studied mainly included moderate cases or organisms with a low MIC, studies have shown that there was no superiority for a CI when compared to an II [37,45]. On the other hand, most studies involving critical patients and/or difficult-to-treat organisms have found a superiority for using a CI. The advantages are more striking when the MICs are higher [28]. 

Besides clinical superiority, a CI has several other advantages. It is associated with less nursing work and enhances patients’ mobility, thus increasing nurses’ and patients’ satisfaction [58]. It is also more cost-effective as CI is associated with less nursing time, and using fewer antibiotics is also associated with lower overall costs [27,52].

CI also has some drawbacks: when multiple medications are given intravenously, a second intravenous line should be inserted to limit the interruptions of antibiotic delivery and limit medication interactions. Keeping an IV line continuously could also limit patients’ mobility; however, the use of an elastomeric pump could incite the patient to ambulate and could also allow for an earlier discharge [59]. Another drawback is that it needs to take into account the stability of different molecules at different temperatures, as imipenem is only stable for 6 h at 25 °C and less at 30 °C; therefore, patients need to be instructed on how to handle the infusion notably in outpatient care [60].

The impact of the CI on the selection of resistant organisms is less clear, as most studies did not report a breakthrough resistance. However, this could be due to the low MICs of the responsible organisms. It remains to be seen if, when treating patients infected with organisms with relatively high MICs, the CI does not prevent the selection of resistance by virtue of a better PK/PD profile. It is unfortunate that very few real-life studies explored this concept of prevention of mutant selection. It will be interesting to see the impact of a generalization of a high-dose CI regimen on the bacterial ecology at a hospital level.

## 9. Conclusions

As evidenced by this review, a one-size-fits-all approach to betalactam therapy might not be a reasonable approach. Physicians and hospital pharmacists should be aware that for severe patients infected with less susceptible organisms, or in the presence of a high bacterial load or a difficult-to-treat infection, it is becoming clearly evident that in order to optimize the clinical response, and due to the absence of a better strategy, a continuous infusion is becoming a “mainstream treatment”. For mild infections, or infections due to a suspected or documented bacteria with low MIC, CI has no gain on clinical response, but it is possible that there is a benefit regarding nursing cost and workload as well as in the emergence of resistance. 

More studies are needed to evaluate CI in implant-associated infections and its positioning in a strategy to combat antibiotic resistance and to evaluate the impact on the selection of resistance.

## Figures and Tables

**Figure 1 antibiotics-12-01052-f001:**
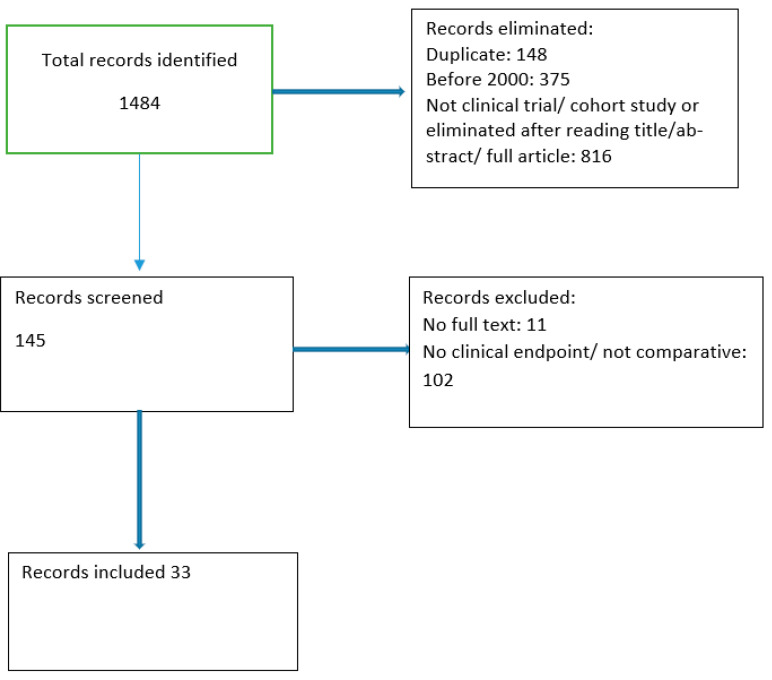
Flow diagram of studies included in the main objective review.

**Table 1 antibiotics-12-01052-t001:** Characteristics of the clinical studies included for the clinical outcome review.

Reference	Study Design	Population, No of Patients	Site of Infection	Antibiotic	Dosage CI	Dosage II
Hughes 2009 [18]	retrospective	Hospital, 107	endocarditis	cloxacillin	12 g over 24 h	2 g q4 h over 30 min
Rafati 2006 [19]	RCT	ICU, 40	various	piperacillin	LD 2 g; 8 g over 24 h	3 g q6 h over 30 min
Laterre 2015 [20]	RCT	ICU, 32	Pneumonia, intra-abdominal	temocillin	LD 2 g, 6 g over 24 h	2 g q8 h over 30 min
Solórzano-Santos, 2019 [21]	RCT	Hospital, 202	Febrile neutropenia	Piperacillin/tazobactam	LD 75 mg/kg; 300 mg/kg/24 h	75 mg/kg q6 h
Cotrina-Luque 2016 [22]	RCT	Hospital, 78	various	Piperacillin/tazobactam	2.25 LD; 9 g over 24 h	4.5 g q8 h over 30 min
Grant 2002 [23]	prospective	hospital, 98	various	Piperacillin/tazobactam	various	various
Lau 2006 [24]	RCT	Hospital, 262	Intra-abdominal	Piperacillin/tazobactam	LD 2.25 g; 13.5 g over 24 h	3.375 g q6 h over 30 min
Dulhunty 2015 [25]	RCT	ICU, 443	various	Meropenem Piperacillin/tazobactam Ticarcillin/clavulanate	3 g over24 h 13.5 g over 24 h 12.4 g over 24 h	1 g q8 h 4.5 g q8 h 3.1 g q6 h Over 30 min
Buck 2005 [26]	prospective	Hospital, 24	CAP/HAP	Piperacillin/tazobactam	2.25 LD; 9 g over 24 h	4.5 g q8 h bolus
De Ryke 2006 [27]	RCT	Hospital, 262	Intra-abdominal	Piperacillin/tazobactam	LD 2.25 g; 13.5 g over 24 h	3.375 q6 h over 30 min
Lorente 2009 [28]	retrospective	ICU, 83	VAP	Piperacillin/tazobactam	18 g over 24 h	4.5 g q6 h over 30 min
Hyun 2022 [29]	retrospective	ICU, 157	sepsis	Piperacillin/tazobactam	LD 4.5, 18 g over 24 h	4.5 g q6 h over 30 min
Li 2010 [30]	RCT	NR, 66	pneumonia	Piperacillin/tazobactam	LD 4.5, 9 g over 24 h	4.5g q8 h over 30 min
Abdul-Aziz (a) 2016 [31]	Post hoc analysis	ICU, 182	various	Meropenem Piperacillin/tazobactam	NR	NR
Abdul-Aziz (b) 2016 [32]	prospective	ICU, 140	various	Meropenem Piperacillin/tazobactam cefepime	LD as II dose; 3 g/24 h, 18 g/24 h, 6 g/24 h	1 g q8 h 4.5 g q6 h 2 g q8 h over 30 min
Dulhunty 2013 [33]	RCT	ICU, 60	various	Meropenem Piperacillin/tazobactam Ticarcillin/clavulanate	Continuous various	Intermittent various
Roberts 2007 [34]	RCT	ICU, 57	sepsis	ceftriaxone	2 g over 24 h	2 g q24 h bolus
McNaab 2001 [35]	RCT	ICU, 41	HAP	ceftazidime	LD 1 g; 3 g over 24h	2 g q8 h over 30 min
Nicolau 2009 [36]	RCT	ICU, 41	HAP	ceftazidime	LD 1 g; 3 g over 24 h	2 g q8 h over 30 min
Hanes 2000 [37]	RCT	ICU, 32	HAP	ceftazidime	LD 2 g, 60 mg/kg over 24 h	2 g q8 h over 30 min
Lorente 2007 [38]	retrospective	ICU, 121	VAP	ceftazidime	LD 1 g, 2 g over 12 h q 12h	2 g q12 h over 30 min
Rappaz, 2000 [39]	Prospective crossover	Hospital, 14	Cystic fibrosis	ceftazidime	100 mg/kg	200 mg/kg divided into 3 doses
Hubert 2009 [40]	Prospective crossover	Hospital, 70	Cystic fibrosis	ceftazidime	LD 60 mg/kg; 200 mg/kg/24 h	200 mg/kg divided into 3 doses
Riethmueller 2009 [41]	Prospective crossover	Hospital, 56	Cystic fibrosis	ceftazidime	100 mg/kg	200 mg/kg divided into 3 doses
Huang 2014 [42]	retrospective	Hospital, 68	Neurosurgical post-op	cefepime	LD 0.5 g,4 g over 24 h	2 g q12 h over 30 min
Georges 2005 [43]	RCT	ICU, 50	HAP, bacteremia	cefepime	4 g over 24 h	2 g q12 h over 30 min
Han 2006 [44]	Pilot study	Hospital, 9	Cystic fibrosis	cefepime	LD 15 mg/kg; 100 mg/kg/24 h	50 mg/kg q8 h over 30 min
Sakka 2007 [45]	RCT	ICU, 20	NR	imipenem	LD 1 g; 2 g over 24 h	1 g q8 h over 40 min
Okimoto 2010 [46]	RCT	Hospital, 50	CAP	meropenem	1 g over 24 h	0.5 g q12 h
Lorente 2006 [47]	Observational, retrospective	ICU, 89	VAP	meropenem	1g q6 h every 6 h	1 g q6 h over 30 min
Zhao 2017 [48]	prospective	ICU, 50	various	meropenem	LD 0.5 g over 30 min; 3 g over 24 h	First dose 1.5 g, then 1 g q8 h
Chytra 2012 [49]	prospective	ICU, 240	various	meropenem	LD 2 g; 4 g over 24 h	2 g q8 h
Helmy, 2015 [50]	prospective	ICU, 100	Severe sepsis	meropenem	LD 2 g, 4 g over 24 h	2 g q8 h over 30 min

## Data Availability

No new data were created.

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
