# Peer review of "Optimizing Betalactam Clinical Response by Using a Continuous Infusion: A Comprehensive Review"

_antibiotics, 2023, doi:10.3390/antibiotics12061052_

Round 1

Reviewer 1 Report (New Reviewer)

Optimizing beta lactam infusion: continuous, extended or intermittent? A comprehensive review. Very good review and certainly needed by clinicians for decision making. Few points to clarify

Generally, all of the studies compared CI vs II were looking at having free drug concentration ? MIC at site of infection. However, PK/PD studies for sites other than blood are lacking (e.g. pneumonia, IAI, UTI, …etc. I suggest that authors comment on this fact

Additionally, most of the published data in critically ill patients did not account for the fact that many of the septic shock patients has compromised renal function and antibiotics dosing in this population should be different – not only for the total daily dosing, but also whether a CI would be different from prolonged (extended) or from II? Authors to comment on this fact as well

Few points similar to the following 

Page 1 “using a continuous perfusion of time-dependent antibiotics” you mean infusion not perfusion?

Author Response

Dear Reviewer,

i agree and it will be changed accordingly.

Reviewer 2 Report (New Reviewer)

The paper "Optimizing betalactam clinical response by using a continuous infusion? A comprehensive review" by Diamantis et al. provides a comprehensive review of the literature on the use of continuous infusion (CI) versus intermittent infusion (II) of betalactams. The authors conducted a thorough search of the literature and included studies that compared the two dosing strategies in a variety of patient populations.

1. One strength of the paper is that the authors provide a clear and concise summary of the existing literature on the topic. They highlight the advantages and disadvantages of CI and II and provide a detailed analysis of the clinical and pharmacoeconomic outcomes associated with each dosing strategy. The authors also provide a clear conclusion based on the available evidence, which is that CI is associated with better clinical response in critically ill patients or those infected by an organism with an elevated MIC.

2. Can you explain why this review is new or telling new things?

3. However, there are some limitations to the paper that should be noted. First, the authors only reviewed articles published in English, which may have excluded relevant studies published in other languages. Second, the authors did not perform a meta-analysis, which could have provided a more comprehensive analysis of the data. Third, the authors did not include studies that compared continuous infusion to other dosing strategies, such as bolus dosing or prolonged infusion. Fourth, the authors did not include studies that evaluated the impact of continuous infusion on the development of antimicrobial resistance. Finally, the authors did not include studies that evaluated the cost-effectiveness of continuous infusion compared to other dosing strategies.

4. The stability of imipenem injection solution was affected by temperature and concentration. Increasing in temperature and drug concentration resulted in decreased stability of imipenem. Suitable temperature and drug concentration should be concerned when this drug is given by extended infusion and continuous infusion. I suggest added more reference and detailed description of stability of imipenem injection solution was affected by temperature and concentration.

Please see (add): https://doi.org/10.1177/15593258211059325

5. Overall, the paper provides a useful summary of the existing literature on the use of continuous infusion versus intermittent infusion of beta-lactams. However, further research is needed to fully evaluate the clinical and economic impact of continuous infusion and to compare it to other dosing strategies.

6. Please provide more data on the importance of physicians and pharmacy around the world to recognize the evidence behind the use of CI of beta-lactams in clinical settings in terms of better clinical outcomes.

Minor editing of English language required.

Author Response

we thank the reviewer for his comments.

we think that it is important that physicians use a continuous infusion for critically ill patients as there is more and more evidence supporting its use, for this reasons we thought to present the data in a concise and clear manner. 

we did not do a metaanalysis since there was a significant heterogeneity in patients selection in the different studies as well as different dosage used, therefore we feel that most of the studies cannot be compared head to head.

we aknowledged in the discussion that we only included studies in English. We did not compare a CI to an extended infusion since extended infusion is not the "traditional" infusion method.  we also acknowledged that in our discussion.

we also added the article suggested by the reviewer on imipenem stability.

we agree that the data on the selection of resistance and the pharmacoeconomic impact  is limited  and there is a need to have more data which we have stated in our article.  

Reviewer 3 Report (New Reviewer)

Thank you for the opportunity to review the manuscript. I think the review is informative and a nice addition of information for clinicians. The authors did acknowledge the advantages of continuous infusion (CI). However, I think that there are other points can be considered related to CI:

·         It takes longer for the patient to stay with the infusion pump (i.e., may refrain the patient staying in bed longer [although may not be true in all cases].)

·         The patient might think that they need to stay in bed during the duration of infusion versus being encouraged to be mobile during hospitalization to decrease VTE and other issues.

·         Longer contact time with the infusion pump.

·         This might interfere with other medications that needed to be co-administer (e.g., timing and convenience).

·         Consider about the compatibility with other co-administration medications

The pharmacoeconomic data in the literature related to this topic is weak and I am unsure if it can translate into any savings for CI versus intermittent infusion. 

Author Response

we would like to thank the reviewer  for his comments.

we agree that the pharmacoeconomic data is weak and we did not suggest to have it as the lone reason to switch for a continuous infusion. however in some settings it could be an added value. 

we agree that the continuous infusion may interfere with the administration of other medications and limit the mobility of the patients in some cases. we will modify our conclusion to reflect on these issues as well.

This manuscript is a resubmission of an earlier submission. The following is a list of the peer review reports and author responses from that submission.

Round 1

Reviewer 1 Report

The goal of this review is to shed light on the existing literature concerning comparing continuous infusion to intermittent infusion of beta lactams. Actually, the current proposal is interesting. Therefore, I recommend that the current study be published after minor revisions as follows:

1-   The authors mentioned that ‘’The association of improved clinical outcomes with prolonged or continuous infusions is unclear’’.  Could the authors discuss the possible mechanisms for these findings?

2-   Please add the possible mechanisms for this ‘’ PK/PD studies showed overwhelmingly that, compared to II, CI of beta-lactam antibiotics is superior for attainment of PK/PD targets’’.

Author Response

we would like to thank the reviewver for his comment and we will adress them:

1: we have rewrote the section involved for clarity, but what was meant was that in this particular metaanalysis they did not find an association between a continuous infusion and a improved survival and not that there was not an association, therefore we could not adress it .

2: we have explained why we have that affirmation in our conclusion by showing what parameter are higher.

Reviewer 2 Report

The manuscript entitled “Optimizing betalactam infusion: continuous, extended or intermittent? A comprehensive review is focused on beta-lactam antibiotics. However, the manuscript can be improved a lot with rewriting of Abstract, Introduction, methodology and conclusion sections. The manuscript title can be improved as it does not focus mainly on optimization.

1. The overall concept of the manuscript is good, but it lacks novelty and general significance. The methodology is missing in the abstract. Clear conclusion is also missing in the abstract.

2. The manuscript should be thoroughly reviewed for grammatical and punctuation errors. Few examples are:

Introduction: second paragraph: “common 3 carbon and 1-nitrogent ring …………..[7]” or “common 3 carbons and 1-nitrogent containing ring …………..[7].

Introduction: Final two classes” should be written as last two classes”……………….. [11, 12].

Introduction: due mainly due to???? …………….. [13].

3. Use either betalactams or beta-lactams or β-lactams , but uniformly one of the form throughout the manuscript.

4. Each antibiotic is mentioned separately, the data lacks the flow and correlation in between.

5. The provided literature is not sufficient to justify the author's claims. Overall, the manuscript will not add a significant contribution to the field.

Author Response

we would like to thank the reviewer for his time , effort as well as his constructive remarks.

we have rewritten the introduction, abstract and conclusion as well as large parts of the article.

1- we updated the methodology and conclusions of the abstract.

2- we have corrected and reviewed the grammar of the article

3- we used betalactams uniformely through the article;

4-we have carefully thought about this remark, however it was a deliberate approach from our side since  we feel that the lack of flow is compensated by the fact that  it would allow the clinician reading this review to check more easily the data of the antibiotic he is interested in insterad of splitting it between PK/PD section and clinical section.

5- we again thank the reviewver for this comment, but we respectfully disagree; there are no recent comprehensive review on the subject combining both clinical and pharmacological studies, and covering most betalactams; although we agree that the results  from the different studies are discordants and the that there was a heterogeneity in the methodologies, the severity of the included patients and dosage of the antibiotics that could explain this fact.  We feel that most studies who selected severe cases or with relatively resistant organisms are in favor of a CI. we do agree however that there are still areas of uncertainty and that more studiers might be required.

Round 2

Reviewer 2 Report

The manuscript entitled “Continuous or extended beta-lactam infusion: A global review is focused on beta-lactam antibiotics. The current form of manuscript contains extracted data from previous publications without data analysis and comparison.

Specific comments:

1. The method of assessing the eligibility of data and analysis of data are missing.

2. Presentation of data in tabular and graphical forms is missing.

3. A strong discussion about the dose optimization, and comparison between data are missing.

4. Authors may follow the given articles for more information and presentation of data.

1. Guilhaumou R, Benaboud S, Bennis Y, Dahyot-Fizelier C, Dailly E, Gandia P, Goutelle S, Lefeuvre S, Mongardon N, Roger C, Scala-Bertola J, Lemaitre F, Garnier M. Optimization of the treatment with beta-lactam antibiotics in critically ill patients-guidelines from the French Society of Pharmacology and Therapeutics (Société Française de Pharmacologie et Thérapeutique-SFPT) and the French Society of Anaesthesia and Intensive Care Medicine (Société Française d'Anesthésie et Réanimation-SFAR). Crit Care. 2019 Mar 29;23(1):104. doi: 10.1186/s13054-019-2378-9. PMID: 30925922; PMCID: PMC6441232.

2. Grupper M, Kuti JL, Nicolau DP. Continuous and Prolonged Intravenous β-Lactam Dosing: Implications for the Clinical Laboratory. Clin Microbiol Rev. 2016 Oct;29(4):759-72. doi: 10.1128/CMR.00022-16. PMID: 27413094; PMCID: PMC5010748.

3. Imburgia TA, Kussin ML. A Review of Extended and Continuous Infusion Beta-Lactams in Pediatric Patients. J Pediatr Pharmacol Ther. 2022;27(3):214-227. doi: 10.5863/1551-6776-27.3.214. Epub 2022 Mar 21. PMID: 35350159; PMCID: PMC8939270.

4. Haseeb A, Faidah HS, Alghamdi S, Alotaibi AF, Elrggal ME, Mahrous AJ, Abuhussain SSA, Obaid NA, Algethamy M, AlQarni A, Khogeer AA, Saleem Z, Iqbal MS, Ashgar SS, Radwan RM, Mutlaq A, Fatani N, Sheikh A. Dose optimization of β-lactams antibiotics in pediatrics and adults: A systematic review. Front Pharmacol. 2022 Sep 21;13:964005. doi: 10.3389/fphar.2022.964005. PMID: 36210807; PMCID: PMC9532942.

Author Response

In accordance with the suggestions of the reviewer we modified our article.

we included a more elaborate methodology, we added a discussion and finally we compiled the main articles reviewed in two tables

Round 3

Reviewer 2 Report

The current form of manuscript does not fulfill the standard criteria of the journal and therefore recommend rejection.

Authors incorporated complete sentences in table 1 and 2 that needs to minimize. The referencing patter in table needs improvement. The method of assessing the eligibility of data and analysis of data are still not scientifically sound. Presentation of data in graphical/schematic plot forms is missing. The current form of data presentation does not look scientifically robust.

Author Response

we would like to thank the reviewer for the time and effort he dedicated to our work.

we presented the data in a matter that we feel would appeal to a clinician and allow him to retrieve the information in an easy way.

we amended the tables as suggested.

we did not do a structured review nor a meta-analysis and we followed the journal instruction for authors who did not required a graph for data selection